# From Assistance to Agency: Rethinking Autonomy and Control in CI/CD Pipelines

## Abstract

AI agents are assuming active roles in Continuous Integration and Continuous Deployment (CI/CD) workflows, yet the research community lacks a shared vocabulary for describing what it means for CI/CD to be *agentic*, how much decision authority is delegated, and where control should reside. This paper argues that the central challenge in agentic CI/CD is not improving task performance but designing *authority transfer*: determining which decisions agents may make, under what constraints, and with what recourse.

To structure this argument, we introduce a distinction between *data-plane authority* (localized interventions such as patch generation and test reruns) and *control-plane authority* (modifications to pipeline configuration, deployment policies, and approval gates). Drawing on research prototypes and industrial platforms, we show that current systems operate mainly at the data plane under *bounded autonomy*, with safety achieved through surrounding governance infrastructure rather than intrinsic agent guarantees. We identify three patterns: constrained autonomy as the dominant design, external governance as the primary safety mechanism, and a widening gap between deployment momentum and evaluation methodology. We propose a research agenda arguing that control-plane safety and governance mechanisms represent the most urgent open problem, followed by formalization of autonomy boundaries, evaluation frameworks, and modeling human-agent coordination.

## CCS Concepts

• **Software and its engineering** → **Software development process management**; • **Computing methodologies** → *Intelligent agents*.

## Keywords

 CI/CD pipelines, DevOps automation, Agentic systems, Autonomous software systems, AI-assisted software engineering

**ACM Reference Format:**
Anonymous Author(s). . From Assistance to Agency: Rethinking Autonomy and Control in CI/CD Pipelines. In . ACM, New York, NY, USA, 5 pages.

## 1 Introduction

Continuous Integration and Continuous Deployment (CI/CD) is central to modern software delivery [11, 29]. AI techniques have long supported tasks such as failure diagnosis and operational monitoring [24, 27], typically as predictive or advisory components embedded within specific pipeline stages. In these systems, decision authority remains with human operators or fixed pipeline logic.

Recent advances in large language model-based systems and software engineering agents [17, 23, 31] have introduced components capable of planning, tool use, and action execution. Industrial platforms now describe agents that observe pipeline state, generate patches or configuration changes, and submit modifications through repository workflows [2, 14–16, 28]. Despite this rapid adoption, the research community lacks a shared vocabulary for characterizing how autonomous these systems are and where decision authority resides.

This paper argues that the central challenge in agentic CI/CD is not improving remediation performance but designing *authority transfer*: determining which decisions agents may make, under what constraints, and with what recourse. To structure this problem, we distinguish between *data-plane authority*—localized interventions within a pipeline execution (e.g., patch generation, test reruns)—and *control-plane authority*—modifications to pipeline configuration, deployment policies, and approval gates. Delegating control-plane authority alters organizational risk boundaries, yet current systems overwhelmingly confine agents to the data plane.

Drawing on research prototypes and industrial platforms, we characterize the current state as *bounded autonomy* and identify three patterns. First, constrained autonomy dominates: agents operate within predefined action spaces, with human approval mediating integration. Second, safety is achieved primarily through surrounding governance infrastructure (e.g., pull requests and policy gates) rather than intrinsic agent guarantees. Third, deployment momentum is outpacing evaluation methodology, and the field lacks benchmarks suited to decision-making agents.

We conclude by outlining a research agenda that prioritizes control-plane safety and governance mechanisms, followed by formalization of autonomy boundaries, evaluation frameworks, and human–agent coordination. Treating authority transfer as a first-class design problem enables more precise reasoning about agentic behavior in CI/CD pipelines.

## 2 Agentic CI/CD in Practice: Examples from Research and Industry

To ground the discussion, we draw on peer-reviewed research and publicly available industrial documentation. Because many recent developments originate from commercial platforms rather than archival publications, both sources help clarify how agentic capabilities are framed and constrained in practice. Our goal is not exhaustive coverage, but a capability-focused set of exemplars that motivates the observations and research agenda that follow.

### 2.1 Research Prototypes

Early examples of agentic behavior in CI/CD appear in automated repair systems. Repairnator continuously monitors CI failures, synthesizes candidate patches, and submits pull requests for human review [26, 30]. R-HERO extends this pattern by incorporating continual learning into patch generation [4]. These systems observe pipeline state, make decisions about remediation strategies, and generate concrete changes. However, integration authority remains mediated by pull request workflows, limiting their autonomy.

More recent LLM-based repair systems exhibit increasing agent-like behavior. RepairAgent [5] autonomously plans and executes

multi-step repair strategies by invoking tools, gathering diagnostic information, and iterating on candidate patches. ChatRepair [33] uses conversational LLM interaction to fix bugs through iterative dialogue with test feedback. These systems demonstrate richer planning and tool-use capabilities than earlier template-based approaches, yet human reviewers retain final say over merging: no generated patch is integrated without explicit approval. BuildSheriff [35] addresses a complementary problem, triaging CI test failures to their root causes, illustrating how diagnostic intelligence can precede and support remediation agents.

Recent work also combines LLM-driven diagnosis with remediation workflows that resemble agent loops. LogSage analyzes CI/CD logs to detect failures and support remediation actions in an industrial setting [34]. Adjacent DevSecOps research such as Auto-Guard frames self-healing as a proactive control layer that can adapt pipeline responses using reinforcement learning [3]. Bouzenia and Pradel [6] present an LLM agent capable of autonomously setting up and executing test suites for arbitrary projects, demonstrating that agentic capabilities are extending beyond patch generation toward broader build and test automation.

A notable gap separates agent behavior at the *development layer* from the *delivery layer*. Li et al. present AIDev, a large-scale study of AI agent activity on GitHub repositories, enabling analysis of agent-authored pull requests and interaction patterns with human contributors [22]. While this work offers empirical foundations for understanding how agents contribute code, it does not address integration, deployment, or control-plane authority within CI/CD pipelines. This gap is significant: it means the community has growing empirical evidence about agent behavior in code contribution but almost none about agent behavior in release governance.

## 2.2 Industrial Systems

Industrial platforms increasingly describe CI/CD functionality using agent-oriented language, and critically, the pattern is consistent across multiple vendors. Major development platforms have each introduced agentic CI/CD capabilities: GitHub integrates coding agents directly into repository and workflow processes, enabling agents to translate human-readable intent into workflow artifacts, perform root-cause analysis, and propose fixes while merge authority remains mediated by review gates [13, 14]; GitLab's Duo agent platform provides a dedicated pipeline-fix flow within merge request workflows [15]; Amazon Q Developer generates code changes triggered by CI signals, routing results through pull request review [2]; and Google's Gemini CLI GitHub Action enables LLM-driven agents to run within GitHub Actions workflows [16].

Specialized vendors exhibit the same pattern. Nx Self-Healing CI, Datadog's Bits AI Dev Agent, and Gitar's autonomous build-fix engine each monitor pipeline outcomes, generate remediation actions, and route proposed changes through pull request mechanisms [12, 28]. Dagger has published architectural blueprints for constructing self-healing pipelines using AI agents within containerized CI environments [9].

These systems exhibit observable properties associated with agency: perception of pipeline signals, decision-making over candidate actions, and execution through artifact generation. However, applying the data-plane and control-plane distinction, we observe that all publicly documented systems concentrate agent behavior at the data plane. Continuous delivery research emphasizes the importance of release governance and deployment discipline [11, 19, 29]. No system we examined delegates control-plane authority (i.e., modification of deployment policies, approval gates, rollback thresholds, or workflow definitions) without human approval. This is a significant finding: despite the marketing language of autonomy, the industry has converged on a conservative authority model in which decision-making and action execution are present but authority remains structurally mediated by repository permissions, approval gates, and governance policies.

## 2.3 Architectural Implications

The bounded autonomy observed across current systems has architectural consequences. CI/CD pipelines are traditionally deterministic execution graphs governed by explicit approval gates and disciplined deployment strategies [11, 29]. Introducing agentic components embeds adaptive decision loops within these structures.

These hybrid structures resemble feedback-loop architectures from self-adaptive and autonomic systems research, particularly the MAPE-K (Monitor-Analyze-Plan-Execute over shared Knowledge) reference model [21, 32]. However, the analogy is limited in important ways. MAPE-K assumes a single managed element with a well-defined adaptation policy, and its control loop is typically designed by the system architect with full authority over the feedback mechanism [8, 18]. In CI/CD pipelines, authority is *socially mediated*: multiple human stakeholders (developers, reviewers, release managers) share governance responsibility, and the "managed element" spans organizational processes, not just technical infrastructure. An agent that modifies a deployment threshold is not simply executing a control loop; it is intervening in a socio-technical governance structure. The data-plane/control-plane distinction captures this difference: data-plane actions resemble classical MAPE-K adaptations within a bounded execution context, while control-plane actions alter the governance structure, which MAPE-K does not capture. As capabilities expand, pipeline architectures should make authority boundaries explicit, distinguishing advisory, semi-autonomous, and control-plane roles. Treating these distinctions as design variables can prevent unintentional escalation of operational authority.

An important open question concerns autonomy escalation. As agents move from data-plane interventions to control-plane modifications, small expansions of authority may compound over time. For example, an agent initially permitted to generate patches may later be allowed to modify workflow definitions or deployment thresholds. Each incremental extension appears locally justified, yet collectively these changes may shift effective release governance from human-centered review toward agent-mediated control. Understanding how such escalation occurs, and how it can be bounded without eliminating useful automation, represents a critical design and policy challenge.

A related concern is authority visibility. As agentic capabilities are introduced incrementally, developers and operators may not have clear mental models of which decisions are being made autonomously, which are advisory, and which remain human-controlled. Prior work in self-adaptive systems emphasizes the importance of explicit feedback loops and observability for maintaining trust and

control [8, 21]. In CI/CD contexts, insufficient visibility into agent decision processes may lead to misplaced trust, delayed human intervention, or incorrect assumptions about compliance boundaries. Making autonomy levels and authority scopes explicit within pipeline tooling may therefore be as important as improving underlying model performance.

A further concern is adversarial robustness. Agents that generate workflow configurations or modify pipeline artifacts introduce a new attack surface for supply chain compromise. An agent-proposed change that passes CI checks but introduces a subtle vulnerability may be more difficult to detect than a conventional malicious commit, because reviewers may place unwarranted trust in agent-generated artifacts [25]. Similarly, prompt injection or manipulation of the signals an agent observes (e.g., crafted log messages or error outputs) could steer remediation behavior in unintended directions. These risks are amplified at the control plane: an agent that modifies deployment policies or approval gates under adversarial influence could compromise release governance at an organizational level. Security implications of agentic CI/CD therefore extend beyond the confidentiality and integrity of individual artifacts to the trustworthiness of the delivery process itself.

## 2.4 Summary

Across the systems we examined, a dominant pattern emerges: semi-autonomous, repair-oriented agents embedded within human-governed workflows. Decision authority is bounded, action spaces are constrained, and final integration or deployment control remains with human stakeholders. We now formalize this and two related patterns as explicit observations.

## 3 Core Observations

Our analysis of research prototypes and industrial systems reveals three consistent patterns that characterize the current state of agentic CI/CD.

## 3.1 Observation 1: Constrained Autonomy Dominates

Although many systems are described as agentic, most operate under tightly bounded autonomy. Research prototypes from Repairnator through RepairAgent and ChatRepair generate patches but defer integration decisions to human reviewers [4, 5, 26, 33]. Industrial tools across multiple vendors similarly route generated changes through pull request or merge request workflows [14, 15, 28].

In practice, these systems exercise delegated authority within predefined action spaces rather than independent control over pipeline orchestration or deployment strategy. As noted earlier, no publicly documented system delegates control-plane authority without human approval. The dominant pattern is semi-autonomous assistance embedded within existing governance structures, mirroring earlier transitions from manual release engineering to automated CI/CD workflows, where automation sped execution without immediately displacing human release authority [1, 29]. Agentic CI/CD appears to be following a similar incremental trajectory, which suggests that the current bounded-autonomy equilibrium is not accidental but reflects a recurring pattern in how organizations absorb automation into governance-sensitive processes.

## 3.2 Observation 2: Governance Carries the Safety Burden

Safety and accountability in current agentic CI/CD systems are achieved primarily through external governance mechanisms (e.g., pull request approval workflows, repository permissions, and policy gates) rather than intrinsic agent guarantees. None of the reviewed systems uses formal verification of agent behavior or built-in safety constraints independent of the surrounding infrastructure.

This reliance on external governance has a critical implication: as agent capabilities expand to include broader planning and artifact modification [17, 23], the safety model depends entirely on the strength and coverage of surrounding controls. Explicit reasoning about authority boundaries and escalation paths is largely absent from current research and industrial documentation, leaving the governance layer as an unexamined single point of trust.

This governance-first safety model echoes patterns observed in DevSecOps integration, where security automation is typically embedded within policy-constrained workflows rather than granted unconstrained authority [3, 24]. The extension of similar constraints to agentic CI/CD suggests that autonomy expansion will likely remain mediated by compliance and risk management requirements.

## 3.3 Observation 3: Evaluation Lags Behind Deployment

Many industrial platforms now offer agentic CI/CD features [14], yet systematic evaluation methods for these systems remain underdeveloped. Traditional CI/CD metrics emphasized in prior surveys, such as build duration, deployment frequency, and failure rate [11, 24, 29], were designed for deterministic automation and do not capture decision quality, constraint adherence, escalation behavior, or long-term governance impact. Existing AIOps and DevOps evaluation frameworks primarily assess anomaly detection accuracy and operational performance gains [24, 27], which are inadequate for agentic systems that autonomously generate and execute actions on repository artifacts or deployments.

The core issue is that evaluation must move beyond task-level correctness toward system-level assessment. An agent may successfully repair a failing build while increasing reviewer burden, triggering unstable rollback cascades, or subtly reshaping governance practices. Such socio-technical effects are not captured in conventional CI/CD benchmarking approaches, and no evaluation framework currently accounts for them in a standardized manner.

These three observations converge on a single conclusion: agentic CI/CD is not only a model-capability question, but a reallocation of decision authority within socio-technical delivery systems. Constrained autonomy persists because governance mechanisms carry the safety burden, and evaluation has not yet caught up to validate alternative designs. Clarifying how authority is bounded, evaluated, and coordinated is therefore the central research challenge.

## 4 Research Agenda

We outline four research directions, ordered by urgency. Control-plane safety is the most pressing because it addresses the highest-risk frontier of capability expansion. The remaining directions provide the conceptual, empirical, and organizational foundations needed to support safe delegation.

## 4.1 Control-Plane Safety and Governance Mechanisms

As noted earlier, current systems avoid control-plane delegation. Expansion into control-plane modification (i.e., altering pipeline configuration, deployment policies, or approval gates) introduces qualitatively different risks, because such changes affect organizational governance rather than individual artifacts. The adversarial concerns are particularly critical at the control plane, where a single compromised modification can affect all subsequent releases.

Research is needed on safety mechanisms tailored to CI/CD governance, including policy-aware agents that reason about their own authority, constrained action synthesis that prevents control-plane modifications outside explicit delegation, and automated rollback strategies with formally specified guarantees. The MAPE-K model offers a basis for feedback-loop design [8, 21], but must be extended to account for socially mediated authority and multi-stakeholder governance in CI/CD pipelines.

## 4.2 Formalizing Autonomy Boundaries

Current systems implement bounded autonomy in an ad hoc manner, typically relying on pull request workflows and repository permissions to constrain agent behavior. A principled framework is needed to specify and verify autonomy boundaries in CI/CD pipelines. Such a framework could build on access control, policy specification, and runtime monitoring [18], extending these ideas to represent graded delegated decision authority rather than binary permission models. Recent work on LLM agent observability [10] suggests that runtime monitoring of agent decisions is feasible and could enable formal boundary enforcement.

One practical direction is to model decision authority, action scopes, and escalation paths explicitly within pipeline architectures. Such models would allow autonomy to be treated as a configurable design parameter rather than an emergent side effect of tooling. The data-plane/control-plane distinction we propose offers a starting vocabulary for such specifications.

## 4.3 Evaluation Frameworks for Agentic CI/CD

Existing CI/CD metrics overlook decision quality, constraint adherence, and governance impact, and the field lacks shared benchmarks for comparing autonomy levels and their associated trade-offs.

Future work should develop reproducible evaluation environments that capture realistic pipeline workflows, approval processes, and failure scenarios. Related efforts exist in adjacent domains: AIOpsLab evaluates AI agents in cloud operations [7], ITBench offers diverse real-world IT automation tasks for agent evaluation [20], and AIDev analyzes agent-authored pull requests in development workflows [22]. However, no comparable standardized environment exists for CI/CD pipelines that captures approval gates, policy constraints, control-plane changes, and human overrides.

Evaluation criteria should extend to include stability, transparency, human override frequency, and long-term effects. A practical entry point is a minimal protocol that separates remediation effectiveness from governance and stability costs. Drawing on CI/CD and DevOps perspectives [24, 29] and AIOps failure-management concerns [27], benchmark suites for agentic CI/CD should report: remediation success (fraction resolved, time-to-repair under fixed budgets),

governance impact (PR iterations, reviewer interventions, override frequency), stability and regression (recurrence and agent-induced regressions), and policy adherence (violations of scoped action constraints, especially control-plane actions). Such reporting enables comparison of bounded-autonomy designs without assuming full autonomy as the target outcome.

## 4.4 Human-Agent Coordination Models

Current systems rely heavily on human-in-the-loop approval mechanisms, yet little research examines how developers interpret, trust, or override agent decisions in CI/CD contexts. As agents assume broader responsibilities, coordination complexity will increase.

During incident response, ambiguous authority between an agent and an on-call engineer can delay mitigation or produce conflicting actions. Research is needed on interaction models that clarify responsibility boundaries, support meaningful explanations, and prevent ambiguous authority in such high-pressure situations. Understanding how agent behavior reshapes collaboration patterns in DevOps teams is essential for responsible deployment. Empirical datasets such as AIDev [22] provide a foundation for studying agent-developer interaction patterns, though analogous datasets capturing CI/CD-specific coordination remain absent.

Advancing agentic CI/CD requires progress across all four directions, with control-plane safety as the enabling prerequisite. Without governance mechanisms that can bound control-plane authority, broader autonomy delegation risks undermining the release discipline that CI/CD was designed to enforce [11, 19].

## 5 Limitations

Our discussion relies partly on publicly available industrial documentation, which may reflect marketing or incomplete disclosure, so we prioritize observable properties over performance claims. The space is evolving quickly and public evidence on large-scale outcomes remains limited, so we emphasize authority and governance structure over long-term operational effectiveness.

## 6 Conclusion

Agentic CI/CD today is best characterized as *bounded autonomy*: agents operate within human-governed approval structures rather than exercising independent control. Across research prototypes and industrial platforms, we observe three recurring patterns: constrained autonomy as the dominant design, safety achieved through external governance infrastructure, and a growing gap between rapid deployment and systematic evaluation. The distinction between *data-plane authority* and *control-plane authority* clarifies the stakes. Current systems remain comparatively safe because they confine agents to the data plane and avoid delegating control-plane authority. As capabilities expand and competitive pressures intensify, this boundary will be increasingly tested. This paper therefore argues that the central research challenge is the design of *authority transfer*. Control-plane safety and governance mechanisms constitute the most urgent priority, followed by formalizing autonomy boundaries, developing evaluation frameworks, and modeling human–agent coordination. Treating authority transfer as a first-class design problem is essential to ensuring that future agentic CI/CD systems are safe, accountable, and empirically grounded.

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
