# OpenReview forum: "From Assistance to Agency: Rethinking Autonomy and Control in CI/CD Pipelines"
_ACM.org/AIWare/2026/Conference — AIware 2026_

### Official Review · Reviewer_XLAF · 2026-03-06

**Rating:** 2
**Confidence:** 3

**Review:**

It's a position paper, not a research paper. The paper observes patterns and proposes agendas but offers no new empirical data, formal models, or prototypes. The three "observations" are essentially well-reasoned claims, not findings in a rigorous sense.

- The strongest aspect of the paper is its clarity of framing.

- The main weakness is that the paper is more assertive than validated. It presents a persuasive argument, but the evidence base is limited. Much of the industrial discussion relies on public documentation, which the paper itself acknowledges may be incomplete or influenced by marketing narratives.

- The paper argues that the main challenge is not “can the agent fix CI failures?” but “who is allowed to decide what in the delivery process?” Essentially the AI agents are redistributing the tasks -- that has been observed by others as well. Going forward, it would be a good idea to look at how that is done.

- The paper’s contribution is mostly conceptual. It does not present: any formal method, benchmark, or provide directions for empirical dataset.

- The methodological framework discussion is not adequate. WSESE 2026 and WSESE 2025 can provide more suggestions for relevant related works.

**Summary:**

The paper argues that as AI agents become embedded in CI/CD pipelines, the field has been asking the wrong question.

Rather than "how well can agents perform tasks?", the right question is "how should decision authority be transferred to agents, and under what constraints?" To structure this reframing, the authors introduce a distinction between data-plane authority (localized actions like patch generation and test reruns) and control-plane authority (modifications to pipeline configuration, deployment policies, and approval gates).

---

> ### Author Response · Authors · 2026-03-20
>
> Thank you for the thoughtful review and for highlighting the clarity of the paper’s framing.
>
> 1. The submission is intended as a vision paper that synthesizes patterns from research prototypes and industrial systems to frame an emerging design problem (authority transfer in agentic CI/CD pipelines) and propose a research agenda. The concepts of **data-plane** and **control-plane** authority are introduced as a vocabulary to allow practitioners and researchers to reason precisely about autonomy levels or governance boundaries.
>
> 2. We agree that the observations are conceptual claims grounded in examples rather than empirically validated findings. As a vision paper, our intent is to identify architectural patterns (e.g., agent-generated patches routed through pull request workflows, approval gates mediating integration) as observable properties that should hold consistently across software projects, which  motivated the research agenda we propose. Regarding the evidence base, the paper explicitly acknowledges the limitation of industrial documentation being incomplete or marketing-influenced, but we will also note that industrial claims reflect publicly available descriptions and may not capture full system behavior.
>
>
> 3. ***“... who is allowed to decide what in the delivery process?”***
> We agree with this interpretation, and this is exactly our reframing. Our paper is not about observing that agents redistribute tasks, but rather making the governance question explicit by giving it an analytical structure through the data-plane/control-plane distinction.
>
> 4. ***“It does not present: any formal method, benchmark, or provide directions for empirical dataset”***
> As a vision paper, we do not present a formal method or benchmark. However, Section 4.3 outlines concrete evaluation criteria and dataset directions, specifically capturing CI/CD governance aspects such as approval gates, policy constraints, and control-plane changes that existing benchmarks overlook. We will improve this point to make these directions more actionable.
>
> 5. Thank you as well for pointing us to WSESE 2025 and WSESE 2026. We will review these proceedings and incorporate relevant related work where appropriate.
>
> We appreciate the reviewer’s comments and will clarify the positioning and methodological framing of the paper in the revision.

---

### Official Review · Reviewer_kqvr · 2026-03-07

**Rating:** 3
**Confidence:** 3

**Review:**

1.	The introduction outlines recent advancements in the use of LLMs and agents in the CI/CD domain. However, the motivation for the study is not sufficiently articulated. The paper mentions a gap in terms of a “lack of shared vocabulary,” but it does not clearly explain what is meant by shared vocabulary in this context. Similarly, the concepts of “authority transfer” and “bounded autonomy” are introduced without sufficient explanation and supporting references, making it difficult for readers to understand the problem the paper aims to address.
2.	The paper also refers to concepts such as data-plane authority and control-plane authority in the Introduction, but these terms are not clearly defined and are not supported with references. Providing definitions and relevant citations would help readers better understand these concepts and how they relate to CI/CD systems.
3.	Section 3.3 states that “three observations converge on a single conclusion,” but the connection between these observations and the stated conclusion is not sufficiently explained.
4.	The paper does not clearly describe the methodology used in the study. It is unclear whether the paper is based on a systematic literature review, an empirical study or other kinds of studies.
5.	The contribution of this study is not clearly described. The paper proposes a new challenge—designing authority transfer—and introduces the concepts of data-plane authority and control-plane authority. However, it is not clear how these concepts can be applied in practice or research.
6.	Overall, it is not clear whether the paper should be considered a position paper, a literature review, or an empirical study. This makes it difficult to evaluate the paper according to the review criteria.

**Summary:**

This paper discusses the current use of agents in CI/CD workflows in both academia and industry, and proposes emerging challenges and a future research agenda for agentic CI/CD systems.

---

> ### Author Response · Authors · 2026-03-20
>
> Thank you for the thoughtful feedback. We appreciate the suggestions regarding clarification of terminology, motivation, and the positioning of the paper.
>
> 1. **Motivation and shared vocabulary *“the motivation for the study is not sufficiently articulated”***.
> The goal of the paper is to highlight that recent LLM-based systems introduce agent-like decision-making into CI/CD workflows, yet the research community lacks a common conceptual vocabulary for describing how much operational authority these systems have and where control resides. Without such vocabulary, practitioners cannot meaningfully compare systems marketed as "agentic" but with fundamentally different human-override mechanisms, and researchers cannot reason precisely about autonomy levels or governance boundaries. By "shared vocabulary," we mean distinctions that make these differences explicit and comparable. We will further clarify this motivation in the introduction.
>
> 2. **Authority transfer and bounded autonomy.**
> Authority transfer refers to delegating operational decisions from human-controlled pipelines to agent systems, for example, allowing an agent to generate remediation patches or rerun failing tests autonomously. **Bounded autonomy** describes the dominant pattern we observe: agents can propose and execute actions within a constrained scope, but integration remains mediated by governance mechanisms such as pull request workflows that require human approval before merging. We will ensure to define both terms explicitly at first use and add supporting references.
>
> 3. **Data-plane and control-plane authority.**
> **Data-plane** authority covers localized actions within a pipeline execution (patch generation, test reruns, artifact modification), whereas **Control-plane** authority covers modifications to the governance structures that shape the pipeline itself (workflow definitions, deployment policies, and approval gates). The distinction matters because the two carry fundamentally different organizational risk profiles. We will clarify this distinction earlier in the paper.
>
> 4. **Connection between observations and conclusion.**
> Section 3 identifies recurring patterns across current systems. The three observations form a logical chain: constrained autonomy persists (Obs. 1) because external governance carries the entire safety burden (Obs. 2), and the challenge to validate alternative designs (Obs. 3) due to lack of evaluation frameworks. Together, they show that the binding constraint in agentic CI/CD is not agent capability but the absence of principled mechanisms for delegating authority safely within socio-technical delivery systems. This framing also provides a structured way for researchers to operationalize such a challenge. We will make this reasoning chain explicit in the revised Section 3.3.
>
> 5. **Methodology.**
> The paper follows a synthesis-based approach that draws on research prototypes and publicly documented industrial systems, focusing on observable architectural and governance properties rather than performance claims. From this evidence base, we identified recurring patterns in how agent capabilities are integrated and constrained within CI/CD workflows, which ground the conceptual distinctions and research agenda we propose. We acknowledge this is not a systematic literature review and will clarify this methodology.
>
> 6. **Contribution.**
> The contribution of the paper is conceptual but operational: we reframe agentic CI/CD as a problem of authority transfer and introduce the data-plane/control-plane distinction as a concrete vocabulary for reasoning about this design space. This framing has direct implications: (a)  gives system designers a classification heuristic (is this agent action data-plane or control-plane?), (b) gives researchers a basis for comparing autonomy designs, and (c) proposes evaluation frameworks with a principled axis along which to measure governance impact. We will clarify this contribution more explicitly in the introduction.
>
> 7. **Paper type.**
> This submission is intended as a vision paper that synthesizes patterns from research prototypes and industrial systems to frame an emerging design problem (authority transfer in agentic CI/CD pipelines) and propose a research agenda. It does not contribute new empirical data or formal models. We will clarify this positioning earlier in the paper (abstract/introduction) so the paper is evaluated against the appropriate criteria.
>
> We appreciate the reviewer highlighting areas where the framing and terminology could be clearer, and we believe the clarifications described above will make the intended contribution and scope of the paper more explicit.

---

### Official Review · Reviewer_zfmF · 2026-03-09

**Rating:** 4
**Confidence:** 3

**Review:**

# Summary of strengths
> 1. The paper addresses an important emerging topic at the intersection of software engineering and AI agents, particularly within CI/CD pipelines.

> 2. The paper synthesizes evidence from both research prototypes and industrial systems, leading to several important observations regarding bounded autonomy and governance-driven safety.

> 3. The proposed directions (e.g., control-plane safety mechanisms, evaluation frameworks, and human-agent coordination) offer meaningful guidance for future work.

# Summary of weaknesses
> 1. The discussion of related systems could be expanded to better reflect recent progress in LLM-based software engineering agents, especially those evaluated on emerging benchmarks such as SWE-bench.

> 2. Some descriptions may conceptual and high-level, and the paper could benefit from concrete fig examples illustrating the study.

# Detailed comments for authors
> 1. Quality: The paper is generally well-structured and clearly articulates its main argument regarding authority transfer in agentic CI/CD systems.

> 2. Clarity: The paper is clearly written and easy to follow. In particular, the paper structure (i.e., from examples to observations and finally to a research agenda) is logical.

> 3. Originality: While prior work has explored AI-assisted CI/CD automation techniques, this paper provides a higher-level perspective on autonomy and governance, which is relatively underexplored.

> 4. Significance: The topic is highly relevant given the rapid adoption of LLM-based agents in software engineering workflows.

# Questions for authors’ response:
> 1. Recently, the SWE-bench benchmark and related issue-resolution tasks have become widely-used for evaluating LLM-based software engineering agents. A number of advanced agent systems have been developed and evaluated in this context. Why are these agents not discussed in the paper? Including them could potentially strengthen the analysis of current agent capabilities and evaluation practices.

**Summary:**

This paper discusses the emerging paradigm of agentic CI/CD systems and argues that the key challenge is not improving agent performance but designing appropriate authority transfer between human developer and AI agents. The paper introduces a conceptual distinction between data-plane authority and control-plane authority to characterize the decision boundaries of agentic CI/CD pipelines. Based on an analysis of both research prototypes and industrial systems, the paper identifies three main observations: (1) constrained autonomy dominates current agentic CI/CD systems, (2) safety is mainly enforced through external governance mechanisms, and (3) evaluation methodologies lag behind real-world deployment. Finally, the paper proposes a research agenda that highlights control-plane safety, autonomy boundary formalization, evaluation frameworks, and human-agent coordination as the key future directions.

---

> ### Author Response · Authors · 2026-03-20
>
> Thank you for this helpful suggestion. We agree that SWE-bench and related benchmarks are important and will add a brief note clarifying their relevance to our scope.
>
> SWE-bench focuses on issue resolution through patch generation in development workflows. In contrast, our paper examines authority transfer in CI/CD, including data-plane actions (e.g., patch generation) and control-plane decisions (e.g., modifying pipeline policies). These governance aspects are not captured by current benchmarks, thus motivating the research agenda in Section 4, particularly the need for evaluation frameworks tailored to agentic CI/CD systems.